# Parallel Evolution of Sex-Linked Genes across XX/XY and ZZ/ZW Sex Chromosome Systems in the Frog *Glandirana rugosa*

**DOI:** 10.3390/genes14020257

**Published:** 2023-01-18

**Authors:** Shuuji Mawaribuchi, Michihiko Ito, Mitsuaki Ogata, Yuri Yoshimura, Ikuo Miura

**Affiliations:** 1Cellular and Molecular Biotechnology Research Institute, National Institute of Advanced Industrial Science and Technology (AIST), Tsukuba 305-8568, Japan; 2Department of Biosciences, School of Science, Kitasato University, Sagamihara 252-0373, Japan; 3Preservation and Research Center, City of Yokohama, 155-1 Asahi Ward, Yokohama 241-0804, Japan; 4Department of Biology, Faculty of Science, Kyushu University, Fukuoka 819-0395, Japan; 5Amphibian Research Center, Hiroshima University, Higashi-Hiroshima 739-8526, Japan; 6Hiroshima University Museum, Higashi-Hiroshima 739-8524, Japan; 7Institute for Applied Ecology, University of Canberra, Canberra, ACT 2617, Australia

**Keywords:** male-biased mutation, *dN/dS* ratio, gene expression, heterogametic sex

## Abstract

Genetic sex-determination features male (XX/XY) or female heterogamety (ZZ/ZW). To identify similarities and differences in the molecular evolution of sex-linked genes between these systems, we directly compared the sex chromosome systems existing in the frog *Glandirana rugosa*. The heteromorphic X/Y and Z/W sex chromosomes were derived from chromosomes 7 (2n = 26). RNA-Seq, de novo assembly, and BLASTP analyses identified 766 sex-linked genes. These genes were classified into three different clusters (XW/YZ, XY/ZW, and XZ/YW) based on sequence identities between the chromosomes, probably reflecting each step of the sex chromosome evolutionary history. The nucleotide substitution per site was significantly higher in the Y- and Z-genes than in the X- and W- genes, indicating male-driven mutation. The ratio of nonsynonymous to synonymous nucleotide substitution rates was higher in the X- and W-genes than in the Y- and Z-genes, with a female bias. Allelic expression in gonad, brain, and muscle was significantly higher in the Y- and W-genes than in the X- and Z-genes, favoring heterogametic sex. The same set of sex-linked genes showed parallel evolution across the two distinct systems. In contrast, the unique genomic region of the sex chromosomes demonstrated a difference between the two systems, with even and extremely high expression ratios of W/Z and Y/X, respectively.

## 1. Introduction

In vertebrates, sex is determined genotypically or environmentally [1,2]. In the genotypic sex-determination system, a sex-determining gene on the sex chromosome triggers the primary formation of the testis or ovary. The heterogametic sex can be male (XY) or female (ZW). In homeothermic vertebrates such as mammals and birds, the heterogametic sex is male and female, respectively. Each system has been highly conserved within its own taxon for hundreds of millions of years [3,4]. Conversely, in poikilothermic vertebrates, the heterogametic sex is different between taxa, species, or even geographic populations within a species, and a transition between the two systems is possible [5,6,7,8]. In both systems, the Y or W chromosome dominates, or a single dose of the Z or X chromosome determines one sex [1]. Here, a basic question arises about the two systems. What are the differences in the evolutionary strategies of the sex chromosomes and sex-linked genes between the two distinct systems? In mammals and birds, the mutation rates of the Y and Z chromosomes are higher than those of the X and W chromosomes, which are male-driven mutations, and nonsynonymous to synonymous substitution rate ratios are higher in the genes of Y and W chromosomes than in the X and Z chromosomes [9,10,11,12,13,14,15]. The XY and ZW systems feature a convergent evolution of the sex-linked genes derived from different origins of the sex chromosomes in the two distantly related species. To uncover similarities and differences in the molecular evolution of sex-linked genes between the two heterogametic sex systems, a direct comparison of the same set of sex-linked genes is necessary. However, this is difficult because the heterogametic sex is usually conserved within a species. Fortunately, there are a few exceptions that share these two systems within a species [16,17,18], one being the Japanese soil frog (*Glandirana rugosa*).

*G. rugosa* comprises six geographic groups based on sex chromosomes, sex-determination systems, and mitochondrial haplotypes: East-Japan, West-Japan, Neo-West-Japan, XY, ZW, and Neo-ZW groups [5,19,20,21]. The East-Japan group has more recently been described as a new species, *G. reliquia* [22]. Here, we designated it to the East-Japan group (*G. reliquia*). The East-Japan (*G. reliquia*), West-Japan, and Neo-West-Japan groups have an XX/XY sex-determination system and homomorphic sex chromosomes. The XY, ZW, and Neo-ZW groups have an XX-XY or ZZ-ZW sex-determination system and heteromorphic sex chromosomes in either sex. The XY and ZW groups share a phylogenetic origin in the past hybridization between the former two groups of West-Japan and East-Japan (*G. reliquia*); the hybridization is estimated to have occurred approximately five million years ago (MYA) [23,24,25], with evolutionarily recent origins of the sex chromosomes. The X, Y, Z, and W sex chromosomes are derived from chromosomes 7 (2n = 26) [24]. The Z and Y chromosomes are subtelocentric, derived from autosome 7 of the West-Japan group, while the W and X chromosomes are metacentric, and derived from autosome 7 of the East-Japan group (*G. reliquia*) through one inversion [23,25] (Figure 1A). The X, Y, Z, and W chromosomes share the same set of genes, providing an opportunity to elucidate the similarities and differences in the molecular evolution of sex-linked genes between the two different systems of XX/XY and ZZ/ZW. In particular, the origin of sex chromosomes is very recent, and the primary stage of the molecular evolution of sex-linked genes is evident, in contrast to those in mammals and birds with old sex chromosomes.

In this study, we identified 766 sex-linked genes by RNA-sequencing (RNA-Seq), de novo assembly, and gene annotation using gonad, brain, and muscle RNAs from male and female individuals of the XY and ZW populations. We then performed evolutionary and allele-specific expression analyses of the X-, Y-, Z-, and W-genes.

## 2. Materials and Methods

### 2.1. Frogs

One male and one female each of the XY and ZW groups in *Glandirana rugosa* were collected from Ichinomiya city, Aichi Prefecture and Suzu city, Ishikawa Prefecture, respectively. The sex of the specimens was determined by inspection of the gonads after euthanasia. Animal care and experimental procedures were approved by the Committee for Ethics in Animal Experimentation at Hiroshima University (Permit Number: G18-2-2).

### 2.2. RNA-Sequencing, De Novo Assembly, and Gene Annotation

Gonads, brains, and muscles from the frogs were cut into small pieces and flash-frozen in liquid nitrogen. Their RNAs were purified using an RNA isolation kit (ISOGEN, Nippon gene, Japan). Paired-end libraries were prepared using a TruSeq Stranded mRNA Library Prep (illumina, CA, USA). Sequencing was performed using Illumina NovaSeq 6000 (illumina, CA, USA). The quality check and adapter trimming were conducted using FastQC v0.11.9 (https://www.bioinformatics.babraham.ac.uk/projects/fastqc/, accessed on 18 January 2020) [26] and Trimmomatic 0.39 (http://www.usadellab.org/cms/?page=trimmomatic, accessed on 5 January 2021) [27], respectively. Subsequently, de novo assembly was performed using Trinity 2.11.0 (https://github.com/trinityrnaseq/, accessed on 18 January 2020) [28,29], and the candidate coding sequences were estimated using TransDecoder 5.5.0 (https://github.com/TransDecoder, accessed on 18 January 2020). Annotation to *Rana temporaria* CDSs (aRanTem1.1, GCF_905171775.1) [30] and ortholog pair identification were performed using BLAST+ 2.2.31 (BLASTP, ≥90% identity, ≥100 aa, https://blast.ncbi.nlm.nih.gov/Blast.cgi?PAGE_TYPE=BlastDocs&DOC_TYPE=Download, accessed on 18 February 2021) [31]. Using the BLASTP analysis, duplicated gene sequences were removed and the longest orthologs per gene were extracted. XX, XY, ZZ, or ZW RNA-sequencing was mapped to the assembled XX or ZZ CDSs using BWA 0.7.17 (https://bio-bwa.sourceforge.net/, accessed on 1 March 2021) [32]. The duplicate reads were removed using Picard 2.25.0 (http://broadinstitute.github.io/picard/, accessed on 1 March 2021). We used GATK4 v4.2.0.0 (https://gatk.broadinstitute.org/hc/en-us/articles/360036194592-Getting-started-with-GATK4, accessed on 1 March 2021) to perform variant calling [33]. The sequences of the X, Y, Z, and W chromosome-linked genes were constructed using the “consensus” command in bcftools [34]. BLASTP analysis of the gene sequences between *G. rugosa* and *R. temporaria* indicated that sex chromosome 7 of *G*. *rugosa* is orthologous to chromosome 9 (autosome) of *R*. *temporaria* (Appendix A). The analysis scheme is shown in Appendix A.

### 2.3. Evolutionary Analyses

Protein-based alignments were performed using TranslatorX v1.1 (http://translatorx.co.uk/, accessed on 27 September 2021) [35] and MUSCLE v3.8.1551 (https://www.drive5.com/muscle/, accessed on 14 October 2021) [36], and then trimAl v.1.2 (http://trimal.cgenomics.org/, accessed on 14 October 2021) was used to eliminate the unaligned (gap) regions [37]. Neighbor joining phylogenetic trees were constructed based on the maximum composite likelihood method using MEGA11 (https://www.megasoftware.net/, accessed on 2 June 2022) [38]. Maximum likelihood phylogenetic trees were also constructed using RAxML-NG 1.0.2 (https://github.com/amkozlov/raxml-ng, accessed on 4 June 2021) [39]. The best-fit models of nucleotide substitution were selected by Modeltest-NG 0.1.6 (https://github.com/ddarriba/modeltest, accessed on 4 June 2021) [40].

*t* value (number of nucleotide substitutions per codon), dN (number of nonsynonymous substitutions per nonsynonymous site), dS (number of synonymous substitutions per synonymous site), and dN/dS ratio were calculated using “codeml” in PAML4.8a (free-ratios model, model = 1; user tree, runmode = 0, http://abacus.gene.ucl.ac.uk/software/paml.html, accessed on 18 January 2020) [41]. The statistical analyses were performed using the Fisher exact test and Wilcoxon–Mann–Whitney tests.

The PROVEAN score was estimated for the impact of mutations on the protein’s functionality using BLAST+ v2.4.0 and BLAST_DB (nr_v4, 2020-02-03) (https://www.jcvi.org/research/provean, accessed on 2 June 2022) [42].

### 2.4. Phylogenetic Analyses of Sex Chromosome-Linked Sox3 and tl Genes

We have previously performed phylogenetic analyses using sex chromosome-linked nucleotide sequences of *sox3* and *tl* genes in the XY (Sekigahara and Hamamatsu cities), ZW (Kanazawa and Niigata cities), West, and East groups [25]. To confirm the quality of the *sox3* and *tl* gene models in this study, we constructed phylogenetic trees based on the constructed gene sequences including the previous dataset [25] by using the neighbor joining (NJ) method (Appendix A). The NJ trees of the two genes showed similar topologies comprising two major clades. One clade contained the genes from autosome 7 of the East-J group (Chiba city), X chromosomes of the XY group (Sekigahara, Hmamatsu, and Ichinomiya cities), and W chromosomes of the ZW group (Kanazawa, Niigata, and Suzu cities). The other clade contained the genes of autosome 7 of the West-J group (Hiroshima city), Y chromosomes of the XY group, and Z chromosomes of the ZW group. These results are consistent with the evolutionary scenario of the sex chromosomes: Y and Z chromosomes were derived from autosome 7 of the West-Japan group, whereas X and W chromosomes were derived from autosome 7 of the East-Japan group (*G. reliquia*) (Figure 1A) [23,25]. It was then confirmed that the gene models constructed in this study have sufficient qualities for analysis on the molecular evolution of the X, Y, Z and W chromosome-linked genes in this species.

### 2.5. Expression Analyses

The analysis of the allele-specific expression between the X and Y chromosome-linked genes in the XY male or Z and W chromosome-linked genes in the ZW female was performed based on each sex chromosome-linked genes using ASE-TIGAR (http://nagasakilab.csml.org/ase-tigar/, accessed on 17 June 2021) [43]. The statistical analysis was performed using Wilcoxon–Mann–Whitney tests.

## 3. Results

### 3.1. Identification of X, Y, Z, and W Sex Chromosomes-Linked Genes

To identify the sex chromosome-linked coding sequence of *G*. *rugosa*, we performed RNA-sequencing, de novo assembly, and gene annotation (see “Section 2”). We then blasted the identified RNA sequences to the *R. temporaria* database and obtained 13,884 and 14,403 genes from the XX and ZZ RNA sequences, respectively. Of them, 12,572 genes were determined to be common orthologs between the XX and ZZ RNAs by BLASTP. Then, we mapped the XX and XY or ZZ and ZW RNA sequences to the 12,572 XX or ZZ genes, respectively. Finally, we constructed the X, Y, Z, and W sex chromosome-linked gene models, which comprised 766 sex chromosome-linked genes, 11,628 autosome-linked genes, and 178 unplaced genes (Appendix A and Appendix A).

To confirm the sex linkage of the 766 sex chromosome-linked genes, concatenated phylogenetic trees of 11,628 autosomal chromosome-linked genes and the 766 genes were constructed by RaxML-NG. The trees displayed their evolutionary scenarios of autosomes and sex chromosomes, respectively (Figure 1A and Appendix A). The bootstrap values were 100 in the autosomal tree, but relatively lower (65%) in the sex chromosomal tree. The findings verified the Y-, X-, Z-, and W-genes as the sex chromosome-linked genes.

**Figure 1 genes-14-00257-f001:**
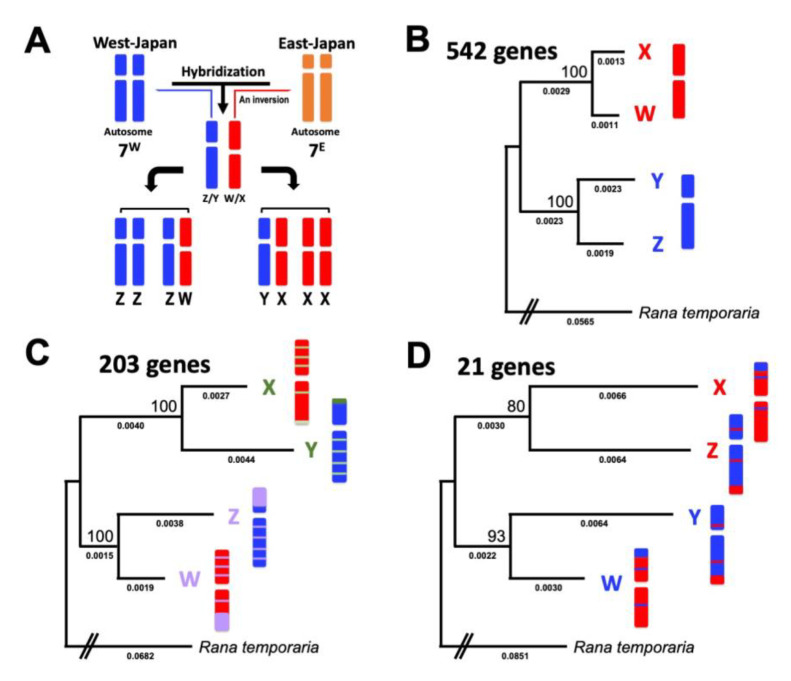
Sex chromosome evolution in *Glandirana rugosa* and sex-linked gene trees. (**A**) Diagrammatic representation of the sex chromosome evolution in *G. rugosa*. The XY and ZW sex chromosomes originated from the hybridization between two ancestral type populations of the West-Japan and East-Japan (*G. reliquia*) groups. X and W chromosomes are indicated in red and the homologous autosome 7 of the East-Japan group (*G. reliquia*) in orange, while Y, Z, and homologous autosome 7 of the West-Japan group are indicated in blue. (**B**–**D**) Phylogenetic trees were constructed by PAML using 542, 203, and 21 sex chromosome-linked genes (free ratio model) from three different clusters, XW/YZ, XY/ZW, and XZ/YW, respectively. The best-fit models of nucleotide substitution were selected by the Modeltest-NG. Numbers at each node denote the bootstrap percentage values based on 1000 replicates. Numbers below the branches are the expected mean numbers of nucleotide substitutions per site. *Rana temporaria* was used as an outgroup.

### 3.2. Three Clusters of Sex-Linked Genes

Based on the topology of the tree of each gene constructed by RaxML-NG, the 766 sex-linked genes were classified into three different clusters: XW/YZ (542 genes), XY/ZW (203 genes), and XZ/YW (21 genes) (Appendix A). Then, the concatenated tree of each of the three clusters was constructed by RaxML-NG (Figure 1B–D). The local bootstrap values in the three gene trees were 100, 100, and 80/93, respectively (Figure 1B–D).

We then investigated the genomic distributions of the genes belonging to the three clusters on chromosome 9 of *R. temporaria*, which is orthologous to the sex chromosomes of *G. rugosa*. The genes of the three clusters were distributed evenly along the chromosomal axis, except for one region spanning from 160 to 184 M (out of 0–184 M), where 84.6% of the genes of the XY/ZW cluster were concentrated, while 10.8% and 4.6% of the genes of the XW/YZ and XZ/YW clusters, respectively, were concentrated (Figure 2). The sequence identity between the Z- and W-genes of the XY/ZW cluster in the 160–184 M region was 99.87% and was significantly higher than 99.29% in the 0–160 M region (Tukey HSD, *p* < 0.001 in Figure 3C). On the other hand, the sequence identities between the X- and Y-genes were 99.22% and 99.47% in the 0–160 M and 160–184 M regions, respectively, which were not significantly different from each other (Turkey HSD, *p* = 0.1274, in Figure 3C). In contrast, the sequence identity between the X- and Y-genes of the XZ/YW cluster was 95.74% in the 160–184 M region and significantly lower than 99.32% in the 0–160 M region (Tukey HSD, *p* < 0.001 in Figure 3D).

### 3.3. Male-Biased Mutation

To elucidate the evolutionary rates of the X-, Y-, Z-, and W-genes, we performed a phylogenetic analysis of the 766 sex-linked genes of three clusters using the PAML free ratio model. We obtained the averaged values of nucleotide substitutions per codon (t) of the sex-linked genes in the branches from the nodes to tips of the X-, Y-, Z-, and W- genes. The nucleotide substitution rates of the Y- and Z-genes were significantly higher than those of the X- and W-genes, respectively (Fisher exact test, *p* < 0.05, XY; *p* < 0.001, ZW) (Figure 4).

### 3.4. Female-Biased dN/dS Ratio

To elucidate the strength of natural selection in the sex-linked genes, we calculated the number of synonymous and nonsynonymous nucleotide substitutions of the X-, Y-, Z-, and W-genes by PAML and then the number of nonsynonymous substitutions per nonsynonymous site (dN), number of synonymous substitutions per synonymous site (dS), and their ratios (dN/dS) of each gene (Figure 4) or concatenated sequences of each of the X-, Y-, Z-, and W-genes (Table 1). The dN/dS_X_ and dN/dS_W_ ratios were higher than the dN/dS_Y_ and dN/dS_Z_ ratios, respectively, indicating female-biased dN/dS ratios (Table 1 and Figure 4).

Subsequently, we performed Gene Ontology enrichment analysis in the four different categories of dN/dS ratios (dN/dS_Y_ > dN/dS_X_, dN/dS_Y_ < dN/dS_X_, dN/dS_Z_ > dN/dS_W_, and dN/dS_Z_ < dN/dS_W_) in the sex chromosome linked genes. The categories did not differ significantly from each other. No selection pressure on gene function at the chromosome scale was observed during sex chromosome evolution in this species.

We also investigated the distribution of t, dN, dS, and dN/dS (X and Z values were subtracted from Y and W values, respectively) of the sex-linked genes along the chromosomal axis and compared the two systems. t, dS, and dN/dS were negatively correlated (R = −0.40, −0.51, and −0.50, respectively), whereas dN was slightly positively correlated with each other (R = 0.17) (Figure 5).

### 3.5. Y- and W-Biased Expressions

To elucidate the allelic expression of the sex-linked genes in different organs of *G. rugosa*, we calculated the allele-specific expression of each of the three clusters between the X, Y, Z, and W chromosomes. Y- and W-biased expression was observed in the XW/YZ and XY/ZW clusters (*p* < 0.05 or *p* < 0.001), but not in the XZ/YW cluster, except in the ZW brain (*p* < 0.01) (Figure 6 and Appendix A).

Next, we investigated the distribution of the Y/X and W/Z expression ratios along the chromosomal axis. The two expression ratios were positively correlated with each other at the whole region from 0–184 M (R = 0.22). Particularly unique was that at the position 160–184 M, the Y/X and W/Z ratios were negatively correlated with each other (R = −0.42) in contrast to that in the 0–160 M region (R = 0.36), where the Y/X ratios were very high, while the W/Z ratios were almost even (Figure 7 and Appendix A).

## 4. Discussion

### 4.1. Three Evolutionary Strata of the Sex Chromosomes in G. rugosa

Based on the topologies of the phylogenetic trees of the sex-linked genes, we identified three clusters: XW/YZ (542 genes), XY/ZW (203 genes), and XZ/YW (21 genes) (Figure 1 and Figure 2). The first cluster, XW/YZ, may have been derived from the autosomal chromosome 7 of the ancestral populations before their hybridization. Thus, this stratum may be the oldest. The XY and ZW sex chromosome systems in *G. rugosa* share a phylogenetic origin of past hybridization between the populations of West-Japan and East-Japan (*G. reliquia*) [5,23,24]. Evidence can be seen from the phylogenetic origins of the sex chromosomes based on the sex-linked gene sequences [23,25]. In this study, we confirmed the dual origin of sex chromosomes (Figure 1B). Based on the gene trees that included orthologous genes of two ancestral populations, it is evident that the Z/Y chromosomes derived from the orthologous autosome 7 of the West-Japan group, while the W/X chromosomes derived from autosome 7 of the East -Japan group (*G. reliquia*) (Appendix A).

The second cluster of XY/ZW may have had two different origins. One is a proto-sex chromosomal origin; the cluster was generated after hybridization between the two ancestral populations and just before or at the sex chromosomal establishment (Appendix A). The two different autosomes 7 might have recombined with each other in the male and female meioses of the hybrids. After the two sex chromosome systems were established and separated into different populations, they started to accumulate independent nucleotide substitutions in each of the two systems. The other is a pseudoautosomal region (PAR) origin generated in the ZW system only (Appendix A). The sequence identities between the Z- and W-genes in the 160–184 M region were significantly higher than those from the other 0–160 M region. Allelic expression of the genes from the 160–184 M region also did not show any W-bias. Thus, the 160–184 M region may be a PAR of the ZW sex chromosomes. Morphologically, the terminal regions of the Z chromosome short arm and W chromosome long arm are estimated to be PARs based on the distribution of chiasmata during female meiosis and the shared signals in the male and female karyotypes by comparative genomic hybridization analysis [44,45]. In contrast, PAR was not identified in the XY system; under a microscope, the X and Y chromosomes were observed to be paired by end-to-end formation with no chiasmata during male meiosis [25]. In addition, in highly evolved frogs such as true frogs including *G. rugosa*, the bivalent chromosomes in male meiosis form a ring-shaped bivalent with no internal chiasma. On the other hand, those in female meiosis form chiasmata along the chromosome axis, except around the centromeric region [46,47,48]. Thus, the PAR of the X and Y chromosomes, if any, may be an extremely tiny terminal tip region and may differ markedly in size from the PAR of the ZW system.

The third cluster of XZ/YW may have been generated by hybridization between the two systems (Appendix A); this stratum may be the youngest. Our previous study showed that XY populations emigrated into the ZW populations after the two systems were established [49], and that the X and Z chromosomes or Y and W chromosomes were recombined with each other because the WY and XZ frogs can be generated by crossing between the frogs from the two systems [50].

The three strata in *G. rugosa* were structurally different from those observed in the sex chromosomes of other vertebrates, in which one stratum is built by chromosomal rearrangement such as inversion, and the gene members of each cluster are arranged in a sequence [51,52,53]. In contrast, in *G. rugosa*, the strata were dispersed entirely on the chromosomes, except for the PAR of the ZW chromosomes, and may have been built on the process of sex chromosome evolution from ancestral autosomes through proto-sex chromosomes to the third phase of inter-population hybridization after sex chromosome establishment [49]. The second and third strata may have formed through recombination between homologous chromosomes.

### 4.2. Male-Driven Mutation

Male-biased mutations in the Y- and Z-borne genes have been observed in many vertebrates and invertebrates [12,54,55,56]. DNA replication errors during germline cell division are a major source of mutations that are transmitted to the next generation. Because male germ cells proliferate more frequently than female germ cells, nucleotide substitution rates tend to be higher in males than in females [57]. Y chromosomes are always present in males, and Z chromosomes are carried during two-thirds of the lives of generations by males [54]. Therefore, Y and Z chromosome-linked genes evolve faster than X and W chromosome-linked genes. In our previous study, a male-driven mutation was shown to work in the XY and ZW systems of *G. rugosa* based on several sex-linked genes [25]. The comprehensive analysis in this study strongly supports the previous results by showing male-biased substitution rates in the XY and ZW systems.

### 4.3. Constraints by Negative Selection on Sex-Linked Genes

The dN/dS ratio is a measure of the strength of natural selection acting on protein-coding genes [41]. The Y and W chromosomal genes show higher dN/dS ratios than the X and Z chromosomal genes in mammals [9,10], lizards [58], *Drosophila* [11,59], *Silene* [13,14], and birds [15]. Many of these are considered to be due to deleterious mutations occurring in the non-recombining regions of the Y and W chromosomes, leading to degeneration, or inversely, in some specialized genes under positive selection [15,60]. In *G. rugosa*, XY and ZW sex chromosomes are heteromorphic in males and females, respectively [18]. Two inversions, one in Z/Y chromosomes and the other in W/X chromosomes, each from primordial autosome 7, created non-recombining regions on the XY and ZW sex chromosomes [5] and the pseudoautosomal regions of the ZW chromosomes were restricted to their terminals [25,45,61]. Evidently, degeneration of the Y and W chromosomes is progressing because artificially constructed YY and WW embryos die of edemata at early developmental stages due to lethal, degenerated genes responsible for development on the Y and W chromosomes [49,62]. Therefore, we expected that the dN/dS ratios would be biased toward the Y- and W-borne genes in this species, as in mammals and birds. However, the dN/dS in all four sex chromosome-linked genes was small and under 1, indicating that purifying selection acts to conserve the functions of the sex-linked genes. In addition, dN/dS was higher in the W-genes (0.4469) than in the Z-genes (0.3473) in the ZW system, and was higher in the X-genes (0.3246) than in the Y-genes (0.2633) in the XY system. Provean scores, which can estimate the type of selection acting on a gene and are lower in deleterious mutations [42], were totally higher in the Z- and X-genes than in the W- and Y-genes (Appendix A). These scores predict less deleterious substitutions or positive selection in Z- and X-genes. A similar case was observed in the stickleback fish *Pungitius pungitius*, in which the sex chromosomes were evolutionarily young and the dN/dS and Provean scores were larger in the X chromosome than in the Y chromosome [63].

### 4.4. Y- and W-Biased Expression

The allele expression was evidently higher in the Y- and W-genes than in the X- and Z-borne genes, respectively, in the two clusters of XW/YZ and XY/ZW of *G. rugosa*. This result is in sharp contrast to other animals where Y- or W-borne genes showed lower expression than their homologues due to the accumulation of deleterious mutations [64,65,66]. Negative correlations between dN/dS ratios and expression levels have been observed [67,68]. We also found that the dN/dS ratios of the W- and Y-genes in *G. rugosa* were negatively correlated between the two systems along the chromosomal axis, while the Y/X and W/Z expression ratios were positively correlated (Figure 5 and Figure 7). These correlation patterns suggest that dN/dS is not related to the allelic expression of sex-linked genes in either system (R = 0.16 in the XY system and R = 0.05 in the ZW system). We are likely to observe sex-linked gene expression patterns favoring heterogametic sex at the primary stage of young sex chromosome evolution. Likewise, the young sex chromosomes of stickleback fish (*Pungitius pubgitius*) showed higher Y/X expression ratios with lower rates of dN/dS in Y-borne genes [63].

Why is the expression higher in Y- and W-borne genes favoring heterogametic sex? As the data shown above are average values, we investigated the relationships between the Provean scores and allelic expression ratios for every gene from the three clusters. Y-genes with lower Provean scores than X-genes were much higher in expression than the X-genes, while the W-genes with higher Provean scores than Z-genes were much higher in expression than the Z-genes, except in the gonads (Appendix A). These results suggest that less deleterious mutations or positive selection to W-genes are involved in the upregulation of the W-genes, but reason for the Y-gene upregulation may differ and is still unclear.

### 4.5. Unique Terminal Region of the Sex Chromosomes

The Y/X and W/Z expression ratios were positively correlated with each other along the chromosomal axis in the 0–160 M region. In contrast, at the 160–184 M terminal region, the two expression ratios were rather negatively correlated (Figure 7). This may be the PAR of the ZW sex chromosomes, as described above, and a tiny terminal PAR in the XY chromosomes. Although the W/Z expression ratios were nearly even, the Y/X expression ratios were remarkably high in this region. The 24 M terminal region may be a critically different region between the XY and ZW sex chromosomes in this species as well as a good genomic region for evolving a male-determining gene with a much higher Y/X expression rate in the XY system. Next, we searched for candidate genes to determine males with higher Y/X ratios among the regions (Table 2), where one candidate was identified, a steroidogenic enzyme, 17β-hydroxysteroid dehydrogenase 1 (Hsd17B8), which catalyzes the production of estradiol for ovarian differentiation. The W/Z expression ratios of the gene in the three tissues were approximately 1, while the Y/X ratios were 2.58–5.16 (Table 2). *HSD18B1*, another gene of this family, has been identified as a female-determining gene in Seriola fishes [69], and *HSD18B8* itself has been isolated as a differentially methylated gene expressed in the gonads of turtles at the temperature sex-determining stages [70]. A truncated form of *Hsd17B8* on the Y chromosome might be involved in the testis determination in *G. rugosa*. Unexpectedly, at almost the terminal tip of the region, we identified another gene, dachshund family transcription factor 2 (Dach2), of which the W/Z expression ratios were extremely high, 50.4–707, while the Y/X ratios were 0.93–2.83. The orthologous gene in humans is reported to be a candidate for premature ovarian failure [71], and thus could be a candidate for determining females in the ZW system of *G. rugosa*.

On the other hand, from the other genomic regions showing higher expression ratios of both Y/X and W/Z along the chromosomal axis, two genes, negative elongation factor complex member E (NELFE) located at the 31 M region and E3 SUMO-protein ligase ZBED1-like located in the 109 M region, showed very high ratios of both Y/X and W/Z. NELFE is expressed in the gonads of *X. tropicalis* [72], and thus may have the potential to regulate both male and female determination in the two systems. Consequently, to examine the sex-determining functions of the genes listed above, it will be necessary to examine their expression in undifferentiated gonads in genetic male and female tadpoles and then perform a functional analysis.

## 5. Conclusions

The frog *G. rugosa* distributed on the Japanese islands is suitable for use in investigating the similarities and differences in the molecular evolution of X, Y, Z, and W chromosome-linked genes derived from the same homologous chromosomes at the early stage of sex chromosome differentiation. We identified three clusters of sex-linked genes that illustrate the evolutionary strata in the history of sex chromosomes. Molecularly, we confirmed male-biased mutation, female-biased dN/dS ratio (all < 1), and Y- and W-gene-biased expression in the sex chromosome-linked genes, showing parallel evolution across the two distinct systems. Importantly, we identified a unique genomic region at the terminal part of the chromosomes, which may be the PAR of the ZW sex chromosomes. If present, the PAR in the XY sex chromosomes was very small and the expression of the Y-genes was much higher than that of the X-genes. This region may represent the genomic and expression differences between the two systems. In future studies, the analyses of X, Y, Z, and W chromosome-linked genes will be extended to orthologous genes that are still autosomal in the two ancestral populations [73]. The results could reveal the dynamics of molecular evolution from autosomes to sex chromosomes and vice versa, associated with sex chromosome turnover.

## Figures and Tables

**Figure 2 genes-14-00257-f002:**
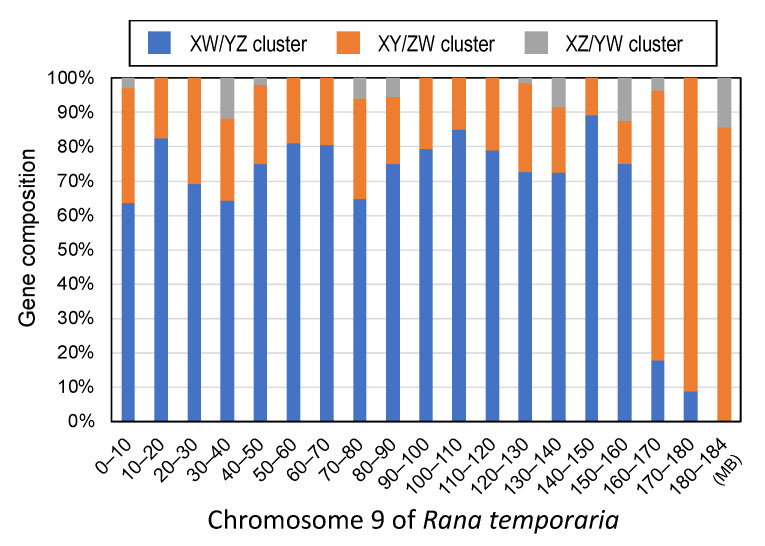
Composition of the sex-linked genes belonging to three different clusters along the chromosomal axis. The genes of the XW/YZ, XY/ZW, and XZ/YW clusters are indicated in blue, orange, and gray, respectively.

**Figure 3 genes-14-00257-f003:**
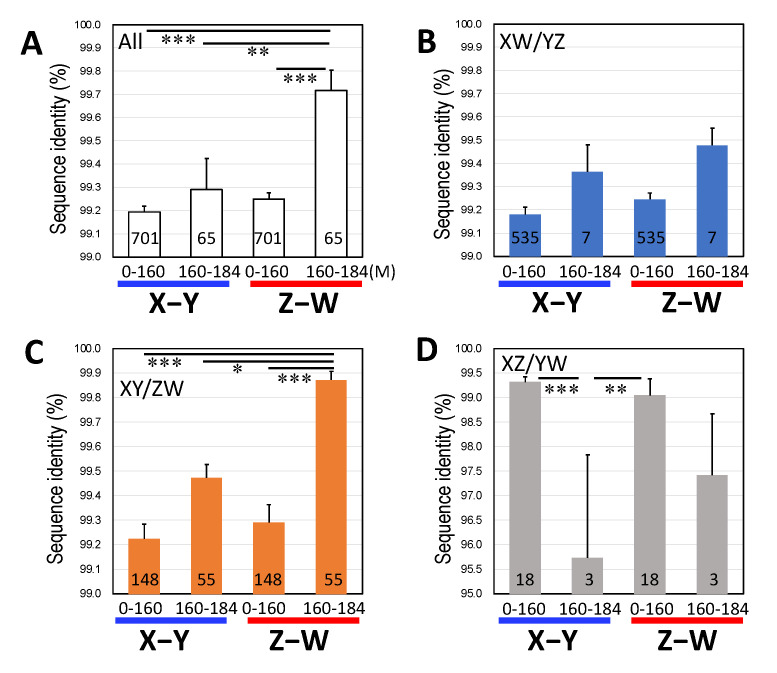
Sequence identities of the sex-linked genes between the sex chromosomes X and Y, and Z and W, located at the regions from 0 to 160 M and 160 to 184 M, respectively. (**A**) All the three clusters. (**B**) XW/YZ cluster. (**C**) XY/ZW cluster. (**D**) XZ/YW cluster. The statistical analysis was performed using Tukey HSD: *, **, and *** indicate *p* < 0.05, *p* < 0.01, and *p* < 0.001, respectively. The numbers of genes used for the analysis are shown on the bars. The abbreviations are the same as in Figure 2.

**Figure 4 genes-14-00257-f004:**
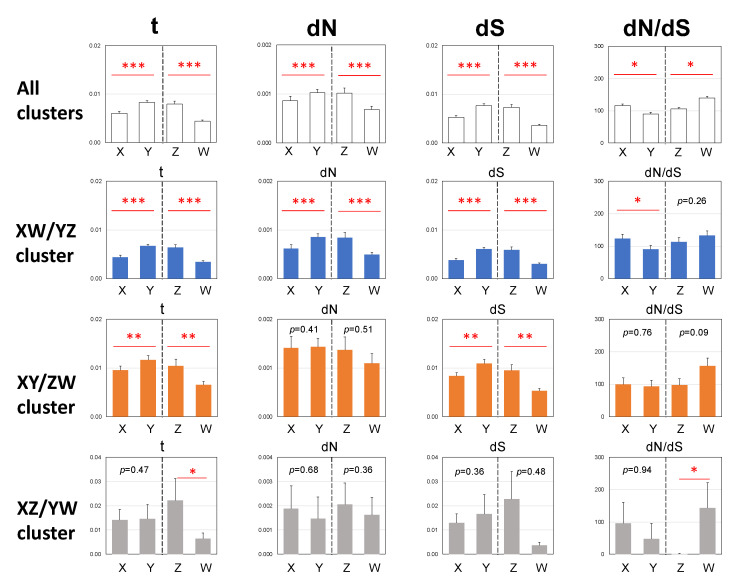
Nucleotide substitutions of the sex chromosome-linked genes. t, number of nucleotide substitutions per codon; dN, number of nonsynonymous substitutions per nonsynonymous site; dS, number of synonymous substitutions per synonymous site; and dN/dS, ratio of dN to dS. *, **, *** are *p* < 0.05, *p* < 0.01, and *p* < 0.001, respectively, by the Wilcoxon–Mann–Whitney tests (paired). The abbreviations are the same as in Figure 2.

**Figure 5 genes-14-00257-f005:**
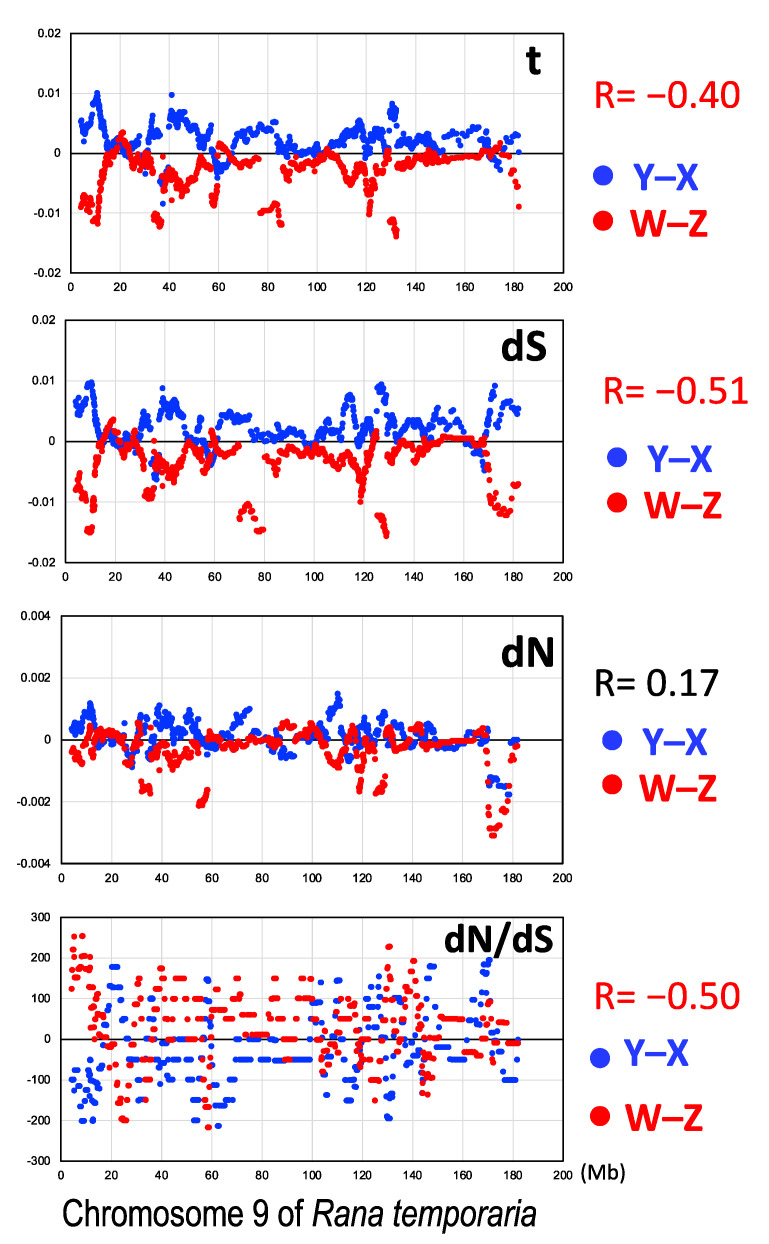
Distribution of the nucleotide substitution rates in the sex-linked genes along the chromosomal axis. Blue circles indicate the subtracted values of the nucleotide substitutions in X-genes from those in Y-genes for t, dS, dN, and dN/dS, while red circles indicate those in Z-genes from those in W-genes. The abbreviations are the same as in Figure 2 and Figure 4.

**Figure 6 genes-14-00257-f006:**
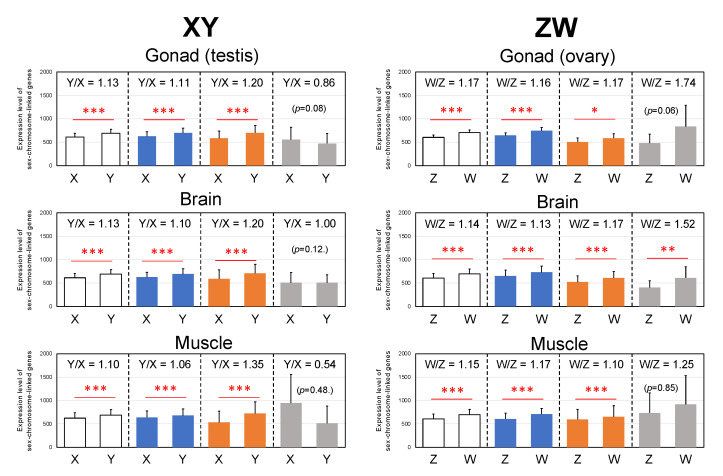
Allelic expression of the sex chromosome linked genes. Statistical analysis of the expression ratio of the sex-linked genes between the sex chromosomes were performed using the Wilcoxon–Mann–Whitney tests (paired). *, **, *** are *p* < 0.05, *p* < 0.01, and *p* < 0.001, respectively. The abbreviations are the same as in Figure 2 and Figure 4.

**Figure 7 genes-14-00257-f007:**
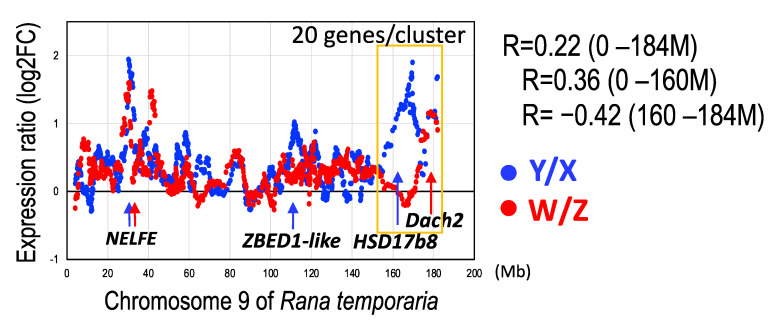
Distribution of the allelic expression ratios of the sex chromosome-linked genes along the chromosomal axis. Blue circles indicate the expression ratios of Y-genes to X-genes, and red circles indicate those of W-genes to Z-genes. The genomic region from 160 to 184 mega bases is boxed in yellow. Arrows indicate the chromosomal positions of the sex determining gene candidates with high expression ratios of Y/X (blue) and/or W/Z (red). The abbreviations are the same as in Figure 4.

**Table 1 genes-14-00257-t001:** Numbers of synonymous and no-synonymous nucleotide substitutions in sex-linked genes.

Cluster	Sex Chromosome	N ^(1)^	S ^(2)^	No. of dN ^(3)^	No. of dS ^(4)^	dN ^(5)^	dS ^(6)^	dN/dS
All	X	866,897.7	316,932.3	646.1 *	1471.4 *	0.0007	0.0046	0.1605
	Y	866,897.7	316,932.3	869.1 *	2304.7 *	0.0010	0.0073	0.1379
	Z	866,897.7	316,932.3	754.5 ***	1976.4 ***	0.0009	0.0062	0.1396
	W	866,897.7	316,932.3	571.2 ***	1155.1 ***	0.0007	0.0036	0.1808
XW/YZ	X	651,460.7	239,629.3	333.8	808.0	0.0005	0.0034	0.1520
	Y	651,460.7	239,629.3	548.8	1430.6	0.0008	0.0060	0.1411
	Z	651,460.7	239,629.3	469.6 *	1230.9 *	0.0007	0.0051	0.1403
	W	651,460.7	239,629.3	323.1 *	706.6 *	0.0005	0.0029	0.1682
XY/ZW	X	199,657.2	71,668.8	265.6 *	575.5 *	0.0013	0.0080	0.1657
	Y	199,657.2	71,668.8	278.5 *	785.2 *	0.0014	0.0110	0.1273
	Z	199,657.2	71,668.8	240.2 **	659.2 **	0.0012	0.0092	0.1308
	W	199,657.2	71,668.8	216.6 **	417.1 **	0.0011	0.0058	0.1864
XZ/YW	X	15,779.8	5634.2	46.7	87.9	0.0030	0.0156	0.1897
	Y	15,779.8	5634.2	41.8	88.9	0.0026	0.0158	0.1679
	Z	15,779.8	5634.2	44.7 *	86.3 *	0.0028	0.0153	0.1849
	W	15,779.8	5634.2	31.5 *	31.4 *	0.0020	0.0056	0.3582

(1) Estimated number of non-synonymous site, (2) Estimated number of synonymous site, (3) Number of nonsynonymous substitutions, (4) Number of synonymous substitutions, (5) Number of nonsynonymous substitutions per nonsynonymous site, (6) Number of synonymous substitutions per synonymous site. Fisher exact test was performed using the rounded values of numbers of nonsynonymous and synonymous substitutions (X versus Y or Z versus W, * *p* < 0.05, ** *p* < 0.01, *** *p* < 0.001).

**Table 2 genes-14-00257-t002:** Four sex-linked genes showing high expression ratios of Y/X and/or W/Z.

			Expression (TPM)		Ratio	Expression (TPM)		Ratio	Chromosomal
ID/Gene Name	Cluster	Tissue	Y	X	Y/X	Z	W	W/Z	Position (bp) ^(1)^
120914319	XY/ZW	gonad	270.5	96.2	**2.81**	252.5	252.5	1.00	168935130
*HSD17B8*		brain	130.5	25.3	**5.16**	66.7	66.7	1.00	
		muscle	158.6	61.3	**2.59**	49.4	49.4	1.00	
120913254	XY/ZW	gonad	2264.9	0.3	**7088.2**	187.0	1956.9	**10.5**	31494656
*NELFE*		brain	1912.9	0.6	**3112.5**	122.7	1486.4	**12.1**	
		muscle	737.3	0.7	**1066.1**	63.1	757.2	**12.0**	
120914576	XY/ZW	gonad	299.5	37.5	**7.98**	3.3	35.2	**10.7**	109121688
LOC120914576		brain	239.9	24.1	**9.96**	34.6	223.0	**6.4**	
*E3 SUMO-protein* *ligaseZBED1-like*		muscle	226.7	10.6	**21.44**	31.9	116.5	**3.6**	
120914364	XY/ZW	gonad	58.6	63.2	0.93	1.1	371.8	**332.9**	178509991
*Dach2*		brain	500.2	231.2	**2.16**	0.8	576.5	**707.4**	
		muscle	52.1	18.4	**2.83**	2.0	103.2	**50.4**	

(1) Chromosome 9 of *Rana temporaria* orthologous to sex chromosome 7 of *Glandirana rugosa*. TPM, transcript per million.

## Data Availability

The assembly of the *Rana temporaria* genomic data is available at https://github.com/DanJeffries/Rana-temporaria-genome-assembly/wiki, accessed on 24 May 2021 [6]. The gene data of *Glandirana rugosa* other than the Appendix A presented in this study are available upon request from the corresponding authors.

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
