# Peer review of "Parallel Evolution of Sex-Linked Genes across XX/XY and ZZ/ZW Sex Chromosome Systems in the Frog Glandirana rugosa"

_genes, 2023, doi:10.3390/genes14020257_

Round 1

Reviewer 1 Report

Decision: I think this is an interesting enough study to be published in Genes. The manuscript is very well-written, and it was a pleasure to review it. I don't have any major complaints, but only minor suggestions.

Minor suggestions:

L26 IMO, there may be a better word than "represented" here.

L31 Perhaps, instead of "following", "reflecting" or "indicating" may be better.

L32 Interesting that the dN/dS for X is higher than that for Y...

L49 The dot is missing...

L53-54 The information on lines 45-46 is repeated, too similar...

L58 By "two cases" you mean the two systems, namely XY and ZW systems, right? Maybe you may change the wording a little?

L69-70 "...we designate it the East-Japan group (G. reliquia) or East-Japan group." IMO, you should improve the wording here and make it more clearer. How about using only one phrase throughout the text?

L72 "XX-XY or ZZ/ZW" Dash or slash?

L77 Again, I'm not sure "represented" is the right word here?

L104 "...libraries were..."?

L116 Of course these are high-quality modern genome sequences, but I'd still liked to see some seq stats like read depth, but this is optional...

L121 "...coding sequence (CDS)..." This abbreviation was already given.

L123-129 Maybe give the numbers on the Results part only, and not here on the M&M? Like "Finally, genes from XX and ZZ RNA sequences were obtained. Common orthologous genes between the XX and ZZ RNAs were determined by BLASTP. Then, we mapped XX and XY or ZZ and ZW RNA sequences to XX or ZZ genes, respectively..." and so on.

L146 "In deleterious mutations, the PROVEAN score is lower [39]." That is right, but maybe it would be better to share this info later, probably in the Results part?

L154 Figure S4: Maybe I missed it, but I guess it is not possible to understand what stands for which gene, using the legend. What do the colors and the depictions of chromosomes represent? Sorry if I missed it...

Another point regarding Figure S4 is that the panel names (i.e., A and B) are apparently misspelled as C and D in the legend. Similarly, the panel names A and B are missing on FigureS2, although it is easy to guess which is which :) On FigureS3, there are two dashes in "sex-linked".

L162-165 This is an important confirmation, even though the bootstrap supports for the branches are relatively low, especially for the gene tl, considering only data from two individuals were used...

L183 "...but relatively lower (65%) in the sex chromosomal tree..."

L189 "...the concatenated tree of each of the three..."

L194 You mean "evenly" instead of "widely" perhaps?

L203-204 The words "identity" and "significantly" are misspelled.

Figure 1 - The numbers on the branches (not the nodes) are not explained in the legend.

L224 I would prefer "...shown on the bars...".

L247-249 So dN/dSX > dN/dSY. This is interesting!

L253-254 Another interesing finding...

L255 The word "and" is misspelled.

Table 1: Footnotes 3 and 4 should be "number of non-synonymous substitutions" and "number of synonymous substitutions", respectively, not per site.

Figure 5 legend: It should be "...substracted values of nucleotide substitutions in X-genes from those in Y-genes for t, dS, dN, and dN/dS, while red circles indicate those in Z-genes from those in W-genes." Right?

L268 Maybe add: "To elucidate the allelic expression of the sex-linked genes in different parts/organs of G. rugosa, we calculated..."

L276 "...in the 0-160 M region (R=0.36), where the Y/X ratios..."

L295-296 I would slightly change the wording like "The first cluster, XW/ZY, may have been derived from the autosomal chromosome 7 of the ancestral populations before their hybridization."

L297-298 "...origin of the past hybridization between the populations West-Japan and East-Japan."

L304 Cool!

L307 Makes sense...

L319 In the legend of FigureS5, it should be stated that the grey stands for PARs.

L323 Maybe some alternative to the phrase "in contrast", which was already used before

L326 "...may differ..."

FigureS6 "...populations have emigrated..."

L349-350 "Z chromosomes are carried during two-thirds of the lives of generations by males." Some reference would be helpful for the readers.

L375-381 I think this is one of the most interesting results of this study. This is a rare observation, but I'm convinced.

L393-400 Kind of a repetition of L371-381. I suggest combining the two main sections somehow, or removing one of these paragraphs.

L407 Typo in "mutations".

Table 2 legend: "TPM stands for transcript per million." Also the word muscle is misspelled.

L413 W/Z...

L434 Another interesting and important finding.

L456 Maybe it would be better to say "The PARs..." instead of "This region..."

Author Response

Dear reviewer 1,

We really appreciate your comments to improve our manuscript. We answered to each of the comments and our responses are written in blue.

Sincerely,

Ikuo Miura and Shuuji Mawaribuchi

-------------------------------

Comments and Suggestions for Authors

Decision: I think this is an interesting enough study to be published in Genes. The manuscript is very well-written, and it was a pleasure to review it. I don't have any major complaints, but only minor suggestions.

Minor suggestions:

L26 IMO, there may be a better word than "represented" here.

We revised as following in L26:

“The heteromorphic X/Y and Z/W sex chromosomes are derived from chromosomes 7”

L31 Perhaps, instead of "following", "reflecting" or "indicating" may be better.

We changed to “indicating” in L31.

L32 Interesting that the dN/dS for X is higher than that for Y...

Thanks.

L49 The dot is missing...

Thanks.  “[5-8].” In L49

L53-54 The information on lines 45-46 is repeated, too similar...

These were summarized to “In mammals and birds” as follows in L53:

“In mammals and birds, the mutation rates of Y and Z chromosomes are higher than those of X and W chromosomes…”

L58 By "two cases" you mean the two systems, namely XY and ZW systems, right? Maybe you may change the wording a little?

Yes, right, and it was revised in L56 to:

“The XY and ZW systems feature convergent evolution …”

L69-70 "...we designate it the East-Japan group (G. reliquia) or East-Japan group." IMO, you should improve the wording here and make it more clearer. How about using only one phrase throughout the text?

Yes, you are right.

We used only one phrase “the East-Japan group (G. reliquia)” through the text, as in L67-68.

L72 "XX-XY or ZZ/ZW" Dash or slash?

Our mistake, thanks.

“XX-XY or ZZ-ZW” in L70.

L77 Again, I'm not sure "represented" is the right word here?

We changed in L75 to:

“The X, Y, Z, and W sex chromosomes are derived from chromosomes 7 (2n=26) [24]

L104 "...libraries were..."?
Thanks:

“Paired-end libraries were prepared ….” In L102.

L116 Of course these are high-quality modern genome sequences, but I'd still liked to see some seq stats like read depth, but this is optional...

We can show the stats based on all transcript contigs of ZZ, ZW, XY and XX of three tissues. The longest orthologs per a gene were used in this study (Figure S1). Please see below. These data do not appear on the manuscript.

*ZZ brain

Trinity stats based on ALL transcript contigs:

Contig N10: 4682; Contig N20: 3270; Contig N30: 2381; Contig N40: 1724; Contig N50: 1177

Median contig length: 376; Average contig: 712.16; Total assembled bases: 314301175

*ZZ muscle

Contig N10: 5217; Contig N20: 3807; Contig N30: 2927; Contig N40: 2272; Contig N50: 1728

Median contig length: 450; Average contig: 913.57; Total assembled bases: 130997993

*ZZ gonad

Contig N10: 5119; Contig N20: 3603; Contig N30: 2645; Contig N40: 1935; Contig N50: 1339

Median contig length: 383; Average contig: 755.98; Total assembled bases: 304701419

*XX brain

Stats based on ALL transcript contigs:

Contig N30: 2522; Contig N40: 1917; Contig N50: 1396

Median contig length: 405; Average contig: 784.49; Total assembled bases: 249649733

*XX muscle

Contig N10: 5481; Contig N20: 3950; Contig N30: 3011; Contig N40: 2337; Contig N50: 1779

Median contig length: 455; Average contig: 928.95; Total assembled bases: 124041651

*XX gonad

Contig N10: 5360; Contig N20: 3942; Contig N30: 3080; Contig N40: 2379; Contig N50: 1783

Median contig length: 401; Average contig: 863.20; Total assembled bases: 161101466

L121 "...coding sequence (CDS)..." This abbreviation was already given.

Thanks. This sentence was removed in the revised manuscript.

L123-129 Maybe give the numbers on the Results part only, and not here on the M&M? Like "Finally, genes from XX and ZZ RNA sequences were obtained. Common orthologous genes between the XX and ZZ RNAs were determined by BLASTP. Then, we mapped XX and XY or ZZ and ZW RNA sequences to XX or ZZ genes, respectively..." and so on.

We modified in M&M section in L99-119 and moved the below part to results section in L163-171:

“To identify sex chromosome-linked coding sequence of G. rugosa, we performed RNA sequencing, de novo assembly, and gene annotation (see “Materials and Methods” section). We then blasted the identified RNA sequences to R. temporaria data base and obtained 13884 and 14403 genes from XX and ZZ RNA sequences, respectively. Of them, 12572 genes were determined to be common orthologs between the XX and ZZ RNAs by BLASTP. Then, we mapped XX and XY or ZZ and ZW RNA sequences to 12572 XX or ZZ genes, respectively. Finally, we constructed X, Y, Z, and W sex chromosomes-linked gene models, which comprised 766 sex chromosome-linked genes, 11628 autosome-linked genes, and 178 unplaced genes (Figure S1 and Table S1)”

L146 "In deleterious mutations, the PROVEAN score is lower [39]." That is right, but maybe it would be better to share this info later, probably in the Results part?

We moved this sentence with modification to Discussion part In L370 because PROVEAN scores first appear in this part: "… are lower in deleterious mutations [42].”

L154 Figure S4: Maybe I missed it, but I guess it is not possible to understand what stands for which gene, using the legend. What do the colors and the depictions of chromosomes represent? Sorry if I missed it...

The sentences were added in Figure S4 legend:

The X- and W-linked genes are boxed in red, the Y- and Z-linked genes and genes on homologous autosomes 7 of West-Japan are in blue, and those on homologous autosome 7 of East-Japan (G. reliquia) are in orange. The X/W, Y/Z chromosomes and autosome 7 of West-Japan, and autosome 7 of East-Japan (G. reliquia) are shown in red, blue, and orange, respectively, and are put on the side of gene tree.

Another point regarding Figure S4 is that the panel names (i.e., A and B) are apparently misspelled as C and D in the legend. Similarly, the panel names A and B are missing on FigureS2, although it is easy to guess which is which :) On FigureS3, there are two dashes in "sex-linked".

Thanks. These were all revised.

L162-165 This is an important confirmation, even though the bootstrap supports for the branches are relatively low, especially for the gene tl, considering only data from two individuals were used...

Yes, we need more data for further confirming the origins of the sex-linked genes.

L183 "...but relatively lower (65%) in the sex chromosomal tree."

It was revised in L176 to: "...but relatively lower (65%) in the sex chromosomal tree..."

L189 "...the concatenated tree of each of the three..."

Revised in L182.

L194 You mean "evenly" instead of "widely" perhaps?

Yes, “evenly” in L187.

L203-204 The words "identity" and "significantly" are misspelled.

Revised in L195 and 197.

Figure 1 - The numbers on the branches (not the nodes) are not explained in the legend.

We added the explanation to legend of Figure 1:

“Numbers below the branches are the expected mean numbers of nucleotide substitutions per site.”

L224 I would prefer "...shown on the bars...".

Changed to: “"...shown on the bars..." in L217.

L247-249 So dN/dSX > dN/dSY. This is interesting!

Thanks.

L253-254 Another interesting finding...
Thanks.

L255 The word "and" is misspelled.

Revised in L248.

Table 1: Footnotes 3 and 4 should be "number of non-synonymous substitutions" and "number of synonymous substitutions", respectively, not per site.

Yes, we revised as below:

3) Number of nonsynonymous substitutions

4) Number of synonymous substitutions

Figure 5 legend: It should be "...substracted values of nucleotide substitutions in X-genes from those in Y-genes for t, dS, dN, and dN/dS, while red circles indicate those in Z-genes from those in W-genes." Right?

Yes, our mistakes. Thanks.

Revised as follows: "...subtracted values of nucleotide substitutions in X-genes from those in Y-genes for t, dS, dN, and dN/dS, while red circles indicate those in Z-genes from those in W-genes."

L268 Maybe add: "To elucidate the allelic expression of the sex-linked genes in different parts/organs of G. rugosa, we calculated..."

It was revised in L261-262 to:

"To elucidate the allelic expression of the sex-linked genes in different organs of G. rugosa

L276 "...in the 0-160 M region (R=0.36), where the Y/X ratios..."

This part was revised in L270 to:
“Particularly unique was that at the position–160-184M, the Y/X and W/Z ratios were negatively correlated with each other (R=−0.42) in contrast to that in the 0-160 M region (R=0.36), where the Y/X ratios were very high, while W/Z ratios were almost even (Figure 7 and Figure S3).

L295-296 I would slightly change the wording like "The first cluster, XW/ZY, may have been derived from the autosomal chromosome 7 of the ancestral populations before their hybridization."

This was revised in L289-290 to “The first cluster, XW/YZ, may have been derived from the autosomal chromosomes 7 of the …”, because this sentence means two autosomes: one is autosome 7 of West-Japan and the other is autosome 7 of the East-Japan (G. reliquia).

L297-298 "...origin of the past hybridization between the populations West-Japan and East-Japan."
It was revised in L291-293 to: The XY and ZW sex chromosome systems in G. rugosa share the phylogenetic origin of the past hybridization between the populations of West-Japan and East-Japan (G. reliquia) [5,23,24].

L304 Cool!
Thanks.

L307 Makes sense...
Thanks.

L319 In the legend of FigureS5, it should be stated that the grey stands for PARs.

This legend was revised to:

“Autosomes 7W and 7E may have been recombined, indicated in grey, after hybridization between the ancestral type-populations of West-Japan and East-Japan (G. reliquia). The genes located on the recombined regions accumulated independently nucleotide substitutions after establishment of the sex chromosomes. The PAR of the terminal regions of Z and W chromosomes are shown in pale purple.”

L323 Maybe some alternative to the phrase "in contrast", which was already used before
It was changed in L317 to: “On the other hand”

L326 "...may differ..."

Revised in L320.

FigureS6 "...populations have emigrated..."

It was revised to:

“XY populations have emigrated into the ZW populations in the past...”

L349-350 "Z chromosomes are carried during two-thirds of the lives of generations by males." Some reference would be helpful for the readers.

The reference [54] was added to the end of the sentence in L344.

L375-381 I think this is one of the most interesting results of this study. This is a rare observation, but I'm convinced.

Thanks.

L393-400 Kind of a repetition of L371-381. I suggest combining the two main sections somehow, or removing one of these paragraphs.
We deleted sentence in L391 (L397-400 in the last text and its reference) because of repetition, but stayed L387-391.

L407 Typo in "mutations".
Revised in L398. And this sentence was modified as follows:

“These results suggest that less deleterious mutations or positive selection to W-genes are involved in the upregulation of the W-genes, but reason for the Y-genes upreguration may differ and is still unclear.”

Table 2 legend: "TPM stands for transcript per million." Also the word muscle is misspelled.

Thanks. Revised.

L413 W/Z...
Revised in L405.

L434 Another interesting and important finding.
Thanks.

L456 Maybe it would be better to say "The PARs..." instead of "This region..."

We stayed the word “this region”, because the region from 160-184M may be PAR of ZW chromosomes, but not of XY chromosomes or it is very much tiny in XY chromosomes.

Submission Date

24 December 2022

Date of this review

09 Jan 2023 18:33:00

Reviewer 2 Report

In the manuscript “Parallel evolution of sex-linked genes across XX-XY and ZZ-ZW sex chromosome systems in the frog Glandirana rugosa”. The manuscript is well-written and the methods are appropriate and well-described. The authors present the molecular evolution of sex-linked genes between XX/XY and ZZ/ZW group of Japanese soil frog (G. rugosa), and can be classified into three evolutionary strata of the sex chromosome. This research is very importance and high impact for evolutionary history of sex chromosome. However, there are require some minor revisions as below;

Line 46 – 47                Please add references for this information.

Line 119                      Please revise “Rana temporaria” to “R. temporaria

Line 193                      Please revise “Rana temporaria” to “R. temporaria

Line 213                      Please revise “XW/YW” to “XW/YZ” group.

Line 219                      Please revise “XZ/YX” to “XZ/YW” clusters. The author should check it carefully for the cluster name.

Line 251                      It might be some mistakes in this comparison. Please confirm the result. It should be “dN/dSW > dN/dSX, dN/dSY < dN/dSX, dN/dSZ > dN/dSY and dN/dSZ < dN/dSW”. Because from Table 1. From all cluster show dN/dS of W > X > Z > Y

Line 270 – 271            For this sentence “Y- and W-biased expression was observed in the XW/ZY and XY/ZW clusters (P<0.01) but not in the XZ/WY cluster, except in the ZW brain”. The P-value should be P < 0.05 because XY/ZW cluster ZW gonad show P < 0.05 not the same as others.

Line 327                      Please add “The third cluster of XZ/WY”. The reader can continuous to understand for this paragraph.

Line 485                      Figure S4. Please revise “Phylogenetic trees of sex chromosome-linked sox3 (C) and tl (D) genes in the XY and ZW groups” to “Phylogenetic trees of sex chromosome-linked sox3 (A) and tl (B) genes in the XY and ZW groups”.

Author Response

Dear reviewer 2,

We really appreciate your comments to improve our manuscript. We answered to each of the comments and our responses are written in blue.

Sincerely,

Ikuo Miura and Shuuji Mawaribuchi

---------------------------------------------

Comments and Suggestions for Authors

In the manuscript “Parallel evolution of sex-linked genes across XX-XY and ZZ-ZW sex chromosome systems in the frog Glandirana rugosa”. The manuscript is well-written and the methods are appropriate and well-described. The authors present the molecular evolution of sex-linked genes between XX/XY and ZZ/ZW group of Japanese soil frog (G. rugosa), and can be classified into three evolutionary strata of the sex chromosome. This research is very importance and high impact for evolutionary history of sex chromosome. However, there are require some minor revisions as below;

Line 46 – 47                Please add references for this information.

Two references were added to L47: O’Meally et al., 2012; Bachtrog et al., 2014.

The reference numbers are 3 and 4.

Line 119                      Please revise “Rana temporaria” to “R. temporaria

Revised in L117.

Line 193                      Please revise “Rana temporaria” to “R. temporaria

Revised in L186.

Line 213                      Please revise “XW/YW” to “XW/YZ” group.

Thanks. Revised in L205.

Line 219                      Please revise “XZ/YX” to “XZ/YW” clusters. The author should check it carefully for the cluster name.

Yes, we again checked it through the text.

Line 251                      It might be some mistakes in this comparison. Please confirm the result. It should be “dN/dSW > dN/dSX, dN/dSY < dN/dSX, dN/dSZ > dN/dSY and dN/dSZ < dN/dSW”. Because from Table 1. From all cluster show dN/dS of W > X > Z > Y

The order does not matter. We performed Gene ontology analysis on the four different categories, and did not find any specific genes to each of the categories. So, we stayed the sentence.

Line 270 – 271            For this sentence “Y- and W-biased expression was observed in the XW/ZY and XY/ZW clusters (P<0.01) but not in the XZ/WY cluster, except in the ZW brain”. The P-value should be P < 0.05 because XY/ZW cluster ZW gonad show P < 0.05 not the same as others.

We revised the sentence in L263-265 as follows:

“Y- and W-biased expression was observed in the XW/YZ and XY/ZW clusters (P<0.05 or P<0.001) but not in the XZ/YW cluster, except in the ZW brain (P<0.01) (Figure 6 and Table S2).

Line 327                      Please add “The third cluster of XZ/WY”. The reader can continuous to understand for this paragraph.

Thanks. Revised in L321 to: “The third cluster of XZ/YW…”

Line 485                      Figure S4. Please revise “Phylogenetic trees of sex chromosome-linked sox3 (C) and tl (D) genes in the XY and ZW groups” to “Phylogenetic trees of sex chromosome-linked sox3 (A) and tl (B) genes in the XY and ZW groups”.

Yes, our mistake.  Revised.

Submission Date

24 December 2022

Date of this review

06 Jan 2023 04:42:16
